# Comparison of Pollutant Effects on Cutaneous Inflammasomes Activation

**DOI:** 10.3390/ijms242316674

**Published:** 2023-11-23

**Authors:** John Ivarsson, Francesca Ferrara, Andrea Vallese, Anna Guiotto, Sante Colella, Alessandra Pecorelli, Giuseppe Valacchi

**Affiliations:** 1Department of Food, Bioprocessing and Nutrition Sciences, Plants for Human Health Institute, NC Research Campus, NC State University, Kannapolis, NC 28081, USA; jcivarss@ncsu.edu; 2Department of Chemical, Pharmaceuticals and Agricultural Sciences, University of Ferrara, 44121 Ferrara, Italy; frrfnc3@unife.it; 3Department of Animal Sciences, Plants for Human Health Institute, NC Research Campus, NC State University, Kannapolis, NC 28081, USA; andrea.vallese@unife.it (A.V.); anna.guiotto@edu.unife.it (A.G.); apecore@ncsu.edu (A.P.); 4Department of Environmental Sciences and Prevention, University of Ferrara, 44121 Ferrara, Italy; 5Department of Neuroscience and Rehabilitation, University of Ferrara, 44121 Ferrara, Italy; 6Department of Biotechnology, Chemistry and Pharmaceutical Sciences, University of Siena, 53100 Siena, Italy; sante.colella@unisi.it; 7Department of Food and Nutrition, Kyung Hee University, Seoul 26723, Republic of Korea

**Keywords:** oxinflammation, inflammasome, pollutants, skin

## Abstract

The skin is the outermost layer of the body and, therefore, is exposed to a variety of stressors, such as environmental pollutants, known to cause oxinflammatory reactions involved in the exacerbation of several skin conditions. Today, inflammasomes are recognized as important modulators of the cutaneous inflammatory status in response to air pollutants and ultraviolet (UV) light exposure. In this study, human skin explants were exposed to the best-recognized air pollutants, such as microplastics (MP), cigarette smoke (CS), diesel engine exhaust (DEE), ozone (O_3_), and UV, for 1 or 4 days, to explore how each pollutant can differently modulate markers of cutaneous oxinflammation. Exposure to environmental pollutants caused an altered oxidative stress response, accompanied by increased DNA damage and signs of premature skin aging. The effect of specific pollutants being able to exert different inflammasomes pathways (NLRP1, NLRP3, NLRP6, and NLRC4) was also investigated in terms of scaffold formation and cell pyroptosis. Among all environmental pollutants, O_3_, MP, and UV represented the main pollutants affecting cutaneous redox homeostasis; of note, the NLRP1 and NLRP6 inflammasomes were the main ones modulated by these outdoor stressors, suggesting their role as possible molecular targets in preventing skin disorders and the inflammaging events associated with environmental pollutant exposure.

## 1. Introduction

Exposure to air pollutants is considered one of the predominant risk factors that can negatively impact human health. One of the primary organs affected by environmental stressors is the skin, which represents the main barrier against the external environment. Due to the skin’s ability to induce inflammatory and oxidative stress reactions throughout the cutaneous tissue, exposure to environmental pollutants has been demonstrated to affect the skin barrier function, altering epidermal proteins and skin lipids and damaging DNA [1,2]. The interplay between oxidative and inflammatory mediators induced by pollutant exposure (oxinflammation) [3] may culminate in the alteration of cutaneous homeostasis, leading to the exacerbation of cutaneous inflammatory disorders (i.e., psoriasis, atopic dermatitis, etc.), as well as skin cancer and premature skin aging [4,5]. One of the main environmental sources of exogenous cutaneous reactive oxygen species (ROS) is represented by UV, which, when absorbed by cutaneous chromophores (e.g., bilirubin, melanin, B6 vitamins, etc.), can promote the production of singlet oxygen (^1^O_2_) and superoxide radical anions (O_2_^•^), which can indirectly damage DNA [6]. Regarding air pollutants, diesel engine exhaust (DEE), one of the principal sources of particulate matter (PM), and cigarette smoke (CS) are recognized to be among the most dangerous to human skin. However, in recent years, other recognized pollutants, such as tropospheric ozone (O_3_) and microplastics (MP), have garnered attention as possible modulators of the inflammatory response in skin disorders [7,8]. The increasing industrialization and use of vehicles has played a major role in increasing the tropospheric O_3_ concentrations in urban areas. Moreover, due to the growing consumption of plastics, MP are now considered a new class of atmospheric contaminants that are dangerous to human health [8]. Although air pollutants can exert similar effects on human skin, their different chemical natures confer upon them different mechanisms of action. For instance, both DEE and CS are composed of a gas phase and a solid phase, but O_3_ or trioxygen can be mainly found in the gas state, and MP are represented by synthetic solid particles. The solid fractions of DEE are mainly composed of PM, a mixture of solid and liquid particles including poly-aromatic hydrocarbons, metals, and inorganic and organic toxins that, when deposited on human skin, are extremely reactive with the cutaneous components of the stratum corneum (cholesterol, fatty acids, and ceramides), triggering oxidative and inflammatory reactions [9]. Moreover, evidence has demonstrated that PM can even penetrate the skin, reaching the deep layers [10], especially when the skin is damaged [11]. CS is a complex aerosol composed of a mixture of more than 4700 chemical substances, among which 1014 are low-molecular-weight carbon- and oxygen-centered radicals [12,13] that can participate in oxidative events [14]. In particular, the main cause of oxidative stress reactions linked to CS is represented by the production of reactive aldehydes during lipid peroxidation events (4-hydroxy-2,3-nonenal and malondialdehyde) [15], or due to the presence of α,β-unsaturated aldehyde species (acrolein and crotonaldehyde) in the CS composition [16]. A similar mechanism of action is ascribed to O_3_, which, although too reactive to penetrate the skin, can directly interact with skin biomolecules, such as lipids, proteins, or DNA, leading to the production of ROS and other nonradical species such as aldehydes (4-hydroxy-nonenal, 4HNE), which can promote the alteration of its structure and function [17,18]. MP, by contrast, originating from the degradation of plastic products, can reach a sub-millimetric size (5 μm or less) and thus be inhaled or ingested [19]. Once introduced into the organism, MP can trigger inflammatory and oxidative stress reactions, along with cytotoxicity and cell death (i.e., apoptosis), due to the presence in their composition of reactive chemicals (i.e., additives, phthalates, heavy metals, etc.) and their ability to carry pathogens [8,20,21]. Recently, it has been found that MP below 100 nm can penetrate human skin. Although the possible mechanism of action of MP within the cutaneous tissue is unclear, it might resemble the behavior displayed by PM [22]. In recent decades, exposure to air pollutants has been shown to modulate cutaneous inflammatory responses via the activation of a new class of inflammatory pathways, namely inflammasomes [23]. Of note, the activation of these multiprotein pathways has been correlated with the exacerbation of different skin conditions, including skin inflammaging [24,25]. The nucleotide-binding and oligomerization domain (Nod)-like receptors (NLRs) represent the most studied inflammasome families involved in the inflammatory response of the innate immune system in many pathological conditions. Upon sensing different stimuli, such as pathogen-associated molecular patterns (PAMPs) or damage-associated molecular patterns (DAMPs), these multiprotein complexes form a platform scaffold due to the assembly of the sensing NLR receptor and the adaptor protein apoptosis-associated speck-like protein containing a CARD (ASC) [26]. The scaffold formation leads to the activation of the caspase-1 enzyme, which can proteolytically cleave pro-inflammatory cytokines IL-1β and IL-18, leading to their maturation and release, therefore promoting an inflammatory status and even a type of cell death called pyroptosis [27]. Among all NLR members (22 up to now), NLRP3, NLRP1, NLRC4, and NLRP6 represent the most studied inflammasomes, involved in many inflammatory tissues pathologies, where NLRP3, NLRP1, and NLRC4 are mainly related to autoinflammatory skin disorders [4]. However, some studies are also exploring the possible involvement of NLRP6 in human skin homeostasis [28]. Although inflammasomes can be activated by a variety of stimuli, ROS have been addressed as one the main insults responsible for inflammasome activation [23].

The aberrant activation of inflammasomes has been correlated to the development of skin conditions such as psoriasis, atopic dermatitis, vitiligo, and many other cutaneous autoinflammatory syndromes [4,29,30,31,32]. Although air pollutants have been shown to differently modulate cutaneous inflammasomes, the role that outdoor stressors can have in the activation of different inflammasomes has never been investigated. Here, we used human skin biopsies to understand how prolonged exposure to air pollutants such as UV radiation, DEE, CS, O_3_, and MP can differently affect specific markers related to oxidative stress, DNA damage, and skin aging. In addition, we explored how long-term exposure to a single environmental pollutant can modulate gene expression, scaffold formation within ASC, the release of inflammatory cytokines, and cell death due to pyroptosis, related to the major NLR inflammasomes (NLRP1, NLRP3, NLR6, and NLRC4) in human skin.

## 2. Results

### 2.1. Effect of Environmental Pollutants on Cutaneous Oxidative Stress and DNA Damage

To evaluate how skin explants’ homeostasis could be affected by different air pollutants, we first exposed the human cutaneous biopsies to MP, CS, DEE, O_3_, and UV for 1 or 4 days and the levels of oxidative stress markers (4-hydroxynonenal (4HNE), Sestrin 2 (SESN2)) and DNA damage (8-oxo-2′-deoxyguanosine (8-OHdG)) were determined. The list of the air pollutants exerting an effect on the abovementioned markers is reported in Appendix A. Human skin biopsies exposed to the different pollutants displayed increased levels of 4HNE at both timepoints, after 1 day (DAY 1) and 4 days (DAY 4) of exposure (Figure 1a). Among all air pollutants, MP and UV seemed to particularly affect the production of 4HNE.

In addition, the levels of Sestrin 2 (SESN2), a highly evolutionarily conserved protein that, in response to various stimuli, controls many cellular responses related to oxidative stress, autophagy, DNA damage, metabolism and inflammation, were also assessed [33]. As depicted in Figure 1b, air pollutants, in particular MP, CS and DEE promoted the decrease of SESN2 levels after the first exposure (DAY 1), and this effect was even more evident at the later timepoint, (DAY 4), where all pollutants promoted a downregulation of SESN2. Since oxidative stress events can lead to DNA damage, the levels of 8-OHdG, one of the major products of DNA oxidation, were also measured. As shown in Figure 1c exposure to air pollutants promoted an increase in 8-OHdG levels, at both timepoints, 1DAY and 4DAY after the exposure. Along with UV radiations, that are known to be one of the main causes of DNA damage within the skin, also MP, CS, DEE and O_3_ significantly promoted oxidation of DNA, especially at DAY 1, suggesting their involvement in a more acute response of the skin against the environmental insults. UV instead, exerted the greatest effect after a prolonged exposure time (DAY 4).

### 2.2. Exposure to Air Pollutants Promotes Extrinsic Aging Markers

The pollutants-induced extrinsic skin aging process [5], was evaluated by measuring the levels of the proteolytic enzyme metalloproteinase-2 (MMP2) and the possible degradation of Type I Collagen. The zymography assay revealed that exposure to the different air pollutants could promote the activation of MMP2 (Figure 2a), one of the main MMPs involved in extracellular matrix (ECM) degradation. Indeed, this technique can identify proteins with gelatinolytic activity based on their capacity to digest the substrate incorporated into the gel. Of note, all air pollutants promoted an increase in active-MMP2 expression levels at the end of the 4 days exposure, with O_3_ and MP displaying the major effect. However, any significant change in MMP2 activation was detected at DAY 1 timepoint. As a result of MMP2 activation, the cutaneous levels of Type I Collagen in response to the different pollutants was also measured (Figure 2b). As depicted in Figure 2b, MP and DEE significantly decreased Type I Collagen levels compared to unexposed tissues already after 1 day of exposure, whereas skin biopsies exposed to the other pollutants did not display any relevant changes although O_3_ and CS showed a trend in the decrease of collagen protein levels. After 4-days exposure, DEE displayed the major effect on Type I Collagen loss, followed by UV and MP.

### 2.3. Air Pollutants Differently Trigger NLRs Inflammasomes Activation in Human Skin Biopsies

Air pollutants have been demonstrated to induce the activation of inflammasomes, master regulators of the inflammatory response of many tissues, including skin [23,24]. Thus, the activation of inflammasomes was evaluated with the aim to understand whether there was a pollutant/NLRs inflammasome specific activation. The list of the air pollutants exerting an effect on inflammasomes components has been reported in Appendix A.

First, it was evaluated the gene expression levels of the main *NLRs* inflammasomes such as *NLRP3, NLRP1, NLRC4* and *NLRP6* in human skin biopsies exposed to the different pollutants (MP, CS, DEE, O_3_ and UV), for 1 (DAY 1) or 4 days (DAY 4). Results showed that beside *NLRC4* (Figure 3c), exposure to the different air pollutants could highly modulate NLRs inflammasomes in human skin. Indeed upon 4 days of exposure, human skin biopsies showed a significant increase in *NLRP3* (Figure 3a), *NLRP1* (Figure 3b) and *NLRP6* (Figure 3d) genes expression compared to unexposed tissue. Among all air pollutants, UV displayed the strongest induction in almost all inflammasomes in response to a prolonged exposure (DAY 4). However, also MP, CS, DEE and O_3_ induced a significant increase in *NLRP3, NLRP1* and *NLRP6* expression levels at DAY 4. *NLRP1* and *NLRP6* turned out to be particularly susceptible to CS and O_3_ at day DAY 1 and upregulated by all air pollutants at DAY 4. MP displayed a tendency to affect *NLRs* inflammasomes induction upon a prolonged air pollutants exposure (DAY 4) rather than a short one (DAY 1). Although DEE exposure could modulate the expression of almost all *NLRs* genes, its effect did not show any tendency towards a specific inflammasome. *NLRC4* turned out to be the less modulated inflammasomes and was upregulated only by CS and UV after one day of exposure. Once activated inflammasomes receptor usually recruit the speck like receptor ASC to create a platform able to eventually promote the release of inflammatory cytokines including IL-1β [26]. Thus, we wondered whether air pollutants could promote the upregulation of ASC. As shown in Figure 3e, beside MP, at DAY 1 and O_3_ at DAY 4, all air pollutants could promote an upregulation of ASC protein expression levels. Of note, UV seemed to have the strongest ability to modulate ASC expression in the short term (DAY 1) compared to the other pollutants, whereas MP, CS and DEE showed a prominent effect in ASC upregulation ad DAY 4.

### 2.4. NLRP1 Most Susceptible Cutaneous Inflammasome to Air Pollutants Exposure

To evaluate the inflammasomes scaffold formation upon air pollutants exposure, the colocalization levels between NLRs inflammasomes and ASC in human skin biopsies were assessed by IF staining. Among all inflammasomes measured, NLRP1 seemed to be the mostly modulated by air pollutants exposure in term of expression levels and colocalization with ASC, whereas NLRC4 turned out to be the less affected. As shown in Figure 4b, almost all air pollutants were able to upregulate NLRP1 expression levels in human skin biopsies at both timepoints, DAY 1 and DAY 4. Of note, the expression levels of ASC were also increased at each timepoint, followed by significant colocalization with NLRP1, especially after 4 days of exposure (DAY 4). These data suggest that NLRP1 might engage the speck-like receptor when activated. NLRP3 expression levels seemed to be upregulated only by DEE exposure after one day (DAY 1), whereas no significant changes in its expression levels were evident after 4 days (DAY 4) of exposure (Figure 4a). However, increased colocalization between NLRP3 and ASC occurred in response to MP and DEE exposure after 4 days (DAY 4).

The NLRC4 and ASC expression levels were upregulated only in response to UV at DAY 1. Exposure to the other air pollutants did not affect the NLRC4 expression levels and in some cases promoted a significant decrease in protein expression levels as well as in the colocalization with ASC (Figure 4c). Regarding NLRP6, its expression level increased after one day of exposure to MP, CS, and O_3_ (DAY 1) and to O_3_ upon 4 days of exposure (DAY 4) (Figure 4d). In this case, only O_3_ exposure promoted an increase in ASC expression levels after 1 day (DAY 1) and 4 days (DAY 4) of exposure. UV, instead, seemed to downregulate NLRP6 expression as well as ASC, especially at DAY 1. Although a tendency towards colocalization between NLRP6 and ASC was visible upon all air pollutants’ exposure, it was not significant compared to unexposed tissues at any timepoint, nor under any conditions.

### 2.5. Air Pollutant Exposure Induced the Activation of Cutaneous Inflammasome and Pyroptosis Responses

To evaluate the effective activation of the inflammatory pathways in response to the different air pollutants, the released levels of IL-1β in the media of the human skin biopsies were measured. As shown in Figure 5a, MP and O_3_ clearly promoted an increase in IL-1β levels after 1 day of exposure (DAY 1), whereas MP, DEE, and UV seemed to induce IL-1β release mainly at longer timepoints (DAY 4) (Figure 5a). Inflammasome activation can eventually lead to a type of cell death named pyroptosis, which includes the formation of pores on the cell membrane that are able to promote the release of inflammatory mediators and propagate the inflammatory response within the tissue, resulting in cell death [34]. This process relies on the substrate Gasdermin D, whose cleavage in response to inflammasome activation generates a 30 kDa N-terminal fragment able to translocate to the cell membrane, where it can form pores [35]. Thus, to determine which air pollutants could eventually be correlated with this type of cell death, the expression levels of active p30 Gasdermin D over the pro-form p50 Gasdermin D in human skin biopsies was assessed by a Western blot. Results showed that after one day of exposure (DAY 1), MP, O_3_, and UV were the main pollutants promoting the cleavage of Gasdermin D in human skin, whereas prolonged exposure resulted in the maturation of Gasdermin D in response to all air pollutants (Figure 5b). In this context, MP, CS, and UV seemed to exert the most evident effect.

## 3. Discussion

The exposure of human skin to environmental pollutants is nowadays recognized as one of the primary causes of the exacerbation of cutaneous disorders, cancer, and premature skin aging [2]. Air pollutants and UV have been shown to induce cutaneous extrinsic premature aging, which itself represents a response to an inflammatory and oxidative stress status (oxinflammation) [36].

For instance, environmental pollutants and UV were found to contribute to the exacerbation of skin photoaging and increased oxinflammatory reactions within the cutaneous tissues, which were counteracted by topical treatment with antioxidant compounds [37]. Clinical studies demonstrated the exacerbation of skin disorders and allergic reactions in subjects living in polluted areas. In Korea, children suffering from moderate to severe atopic dermatitis showed exacerbation of the pathology when exposed to indoor pollutants such as volatile organic compounds (VOCs) [38], and exposure to road traffic increased the allergic response in the skin of patients suffering from topic eczema/dermatitis compared to normal subjects [39]. In addition, PM2.5 exposure levels were positively associated with skin aging manifestation in the Chinese population [40].

The mechanism by which outdoor stressors affect target tissues, including the skin, is mainly through the alteration of the cellular redox homeostasis [41], leading to a cascade of oxidative stress reactions involving the depletion of skin surface antioxidants, cutaneous structural damage, and the impairment of barrier function [2]. All these events can culminate in the activation of inflammatory pathways, among which inflammasomes are now recognized as important modulators of the cutaneous inflammatory response to environmental insults [24].

Therefore, the aim of the present study was to understand how different air pollutants can promote the establishment of an oxidative and inflammatory environment. To do so, human skin biopsies were exposed for 1 or 4 days to different environmental agents such as MP, CS, DEE, O_3_, and UV. Our results confirmed the presence of an oxidative stress environment in response to pollutant exposure, accompanied by increased DNA damage and the initiation of the skin aging process. We then focused our attention on the possible involvement of the main NLR inflammasomes (NLRP1, NLRP3, NLRP6, NLRC4) in the cutaneous response to environmental insults.

First, our results showed that exposure to air pollutants, particularly to MP and UV, promoted increased levels of 4HNE (Figure 1a), one of the most reactive products of lipid peroxidation. Since air pollutants can interact with the lipids present within the *stratum corneum*, leading to the production of ROS and consequent lipid peroxidation, 4HNE represents an important oxidative stress marker in skin conditions [15,42]. Moreover, 4HNE is involved in the regulation of several inflammatory and oxidative stress pathways, including the activation of redox-sensitive transcription factors such as NF-κB and NRF2. In addition, as an α-β unsaturated aldehyde, it is able to covalently react with proteins and form protein adducts (PAs), an event that can lead to changes in protein conformation and thereby in the alteration of their function, eventually resulting in cell death [43,44,45,46,47,48]. Many studies have shown the presence of higher 4HNE levels after O_3_ exposure in several skin models, including in vitro, ex vivo (human explants), and in vivo (human and mouse models) [49]. Other pollutants, such as CS and PM, have also been associated with increased 4HNE PA levels in human skin and keratinocytes [9,50]. Along with the increased 4HNE levels, we also found altered expression of Sestrin 2 (SESN2) (Figure 1b), an evolutionarily conserved protein known to regulate the cellular antioxidant response and the aging process and play a role in tumor suppression, such as melanoma [51]. For example, the knockdown of *SESN2* in melanoma cells was followed by elevated oxidative stress levels and increased apoptosis rates, whereas the upregulation of the *SESN2* gene exerted a protective role by detoxifying oxidative stress species and disrupting subsequent melanoma formation [52]. Our results show that exposure to air pollutants can significantly decrease SESN2 levels even after one day (DAY 1) of exposure, and this effect was even more evident after 4 days of exposure (DAY 4), suggesting the compromised activity of the cellular defensive response. Air pollutants are indeed known for modulating the detoxifying responses of the skin, such as the aryl hydrocarbon receptor (AHR) and nuclear factor erythroid 2-related factor 2 (NRF2) transcription factors, which regulate cutaneous homeostasis [53]. Considering that NRF2 can bind to the antioxidant response element (ARE) within the SESN2 gene, promoting its expression to exert cytoprotective effects against oxidative stress [54,55], our data suggest that air pollutants may alter the antioxidant response of the cutaneous tissue, resulting in an inadequate tissue response to fight the increased oxidative damage.

Excessive ROS production can cause DNA damage via an indirect mechanism that leads to the formation of DNA oxidation products such as 8-OHdG [56]. All air pollutants induced an increase in 8-OHdG expression levels after one day and four days of exposure (Figure 1c). Among all air pollutants, the prolonged exposure to UV (DAY 4) seemed to particularly promote the oxidation of DNA, in line with the evidence that sun radiation represents the main cause of DNA mutation and skin cancer progression [57].

Upon ROS formation, skin cells can release a variety of pro-inflammatory mediators, resulting in an inflammatory status with activated neutrophils and immune cell infiltration, which can generate further free radicals, establishing a vicious cycle named oxinflammation [3]. The continuous secretion of pro-inflammatory mediators (e.g., cytokines, chemokines) can promote the degradation of the connective tissue of the dermis by inducing the activation of matrix metalloproteinases (MMPs). These enzymes can target the degradation of structural proteins of the extracellular matrix (ECM) as different types of collagens, elastins, etc., ultimately affecting the dermis and its functionality and the formation of wrinkles (skin aging). Besides UV, exposure to various air pollutants can also contribute to this process by inducing the redox activation of MMPs, resulting in ECM matrix component degradation [36]. For instance, PM, one of the main components of DEE, has been shown to promote the induction of several MMPs, such as MMP-1, MMP-2, and MMP-9, in human keratinocytes via the ROS/JNK pathway, and treatment with natural antioxidant compounds such as hesperidin was able to partially prevent PM-induced skin damage [58]. Hairless mice exposed to O_3_ displayed increased levels of 4HNE, accompanied by an increase in MMP-9 mRNA and activity levels [59]. Similarly, our results indicated that the prolonged exposure to air pollutants promoted an increase in active MMP-2 in skin biopsies (Figure 2a), especially in response to UV. In line with this result, pro-inflammatory markers as well as the induction of proteolytic enzymes such as MMPs in response to UV have been shown to favor the formation of a microenvironment more suitable for tumor progression and metastasis [60,61]. In parallel to MMP-2 induction, we also found decreased levels of type I collagen, especially at DAY 4, where DEE and UV represented the most dangerous insults (Figure 2b). It is worth mentioning that among all the environmental insults to which the skin biopsies were exposed, MP was found to be one of the major air pollutants triggering oxidative stress (4HNE, SESN2), DNA damage (8-OHdG), and skin aging (MMP2, type I collagen). Considering that the major entry routs for MP are the digestive and respiratory systems, the direct impact that MP can exert on human skin is still poorly understood and under investigation. The presence of the *stratum corneum* in the outermost layer of the epidermis makes cutaneous MP absorption difficult, resulting in MP penetration mainly when the skin is damaged and it is easier to pass through (through wounds, hair follicles, etc.). Our results demonstrate, for the first time, that MP has a direct impact on human skin by altering skin redox homeostasis and even promoting DNA damage. Apart from the direct damage that MP can exert while interacting with human skin, it is possible to hypothesize that inhaled or ingested MP can reach the skin from the bloodstream, making this relatively “new” pollutant one of the most toxic to the cutaneous tissue.

The activation of the inflammatory pathways known as inflammasomes has garnered immense attention in the past few decades in terms of mediating the inflammatory response related to cutaneous disorders [24]. For instance, genetic variations of the NLRP1 and NLRP3 inflammasomes have been associated with high levels of IL-1β in the lesions of patients suffering from psoriasis, vitiligo, and atopic dermatitis [30,32,62,63,64,65,66,67,68,69,70]. Increasing evidence in the past few decades has shown that air pollutants can trigger inflammasome activation in human skin via ROS production [24]. As an example, NLRP1 function was found to be compromised in human keratinocytes upon PM exposure via the ROS/NFkB pathway. Moreover, numerous pieces of evidence correlate UV exposure with the activation of cutaneous inflammasomes, especially via DNA damage and ROS production and subsequent IL-1β and IL-18 release [24,71]. Exposure to O_3_ was found to promote cutaneous NLRP1 inflammasome activation via oxidative stress mediators such as 4HNE and H_2_O_2_; similarly, CS was found to differently modulate the NLRP3 and NLRP1 inflammasomes in human keratinocytes [72]. Although NLRP1 and NLRP3 represent the most studied inflammasomes in human skin, there is some evidence suggesting that cutaneous NLRC4 can be activated by UV [73].

While the number of studies correlating the exposure of air pollutants to cutaneous inflammasome activation has tremendously increased in the past few years, limited information is available on how the same inflammasome receptor can differently respond to specific pollutants.

In the present study, we found that NLRP1 may represent the major inflammasome involved in the cutaneous inflammatory response against environmental insult. The results showed that the *NLRP3* mRNA expression levels were upregulated by all air pollutants, whereas *NLRP1* and *NLRP6* were particularly susceptible to O_3_ and CS at DAY 1 and to all air pollutants upon prolonged exposure (DAY 4) (Figure 3a–d). The *NLRC4* inflammasome was the least upregulated by the environmental insults, only responding to UV (Figure 3c). Moreover, among all the inflammasomes, only NLRP1 seemed to colocalize with ASC, already at DAY 1 but more evidently at DAY 4, suggesting that NLRP1 may represent the inflammasome that is most involved in the cutaneous inflammatory response triggered by environmental insults (Figure 4b). These data are in alignment with the evidence that NLRP1 is the most expressed NLR receptor in human skin [74]. Following NLRP1, NLRP6 also seemed to be particularly susceptible to air pollutant exposure. Indeed, although the colocalization with ASC did not occur properly, the air pollutants increased the NLRP6 expression levels both at DAY 1 and DAY 4 (Figure 4d). Regarding the other inflammasomes, NLRC4 seemed to be the least modulated one in human skin (Figure 4c), confirming the mRNA expression level results. However, in accordance with other studies conducted on NLRC4, UV was found to be the only environmental stimulus able to modulate the NLRC4 inflammasome both at the gene and protein expression levels [73].

Although we found that NLRP3 was the major inflammasome modulated at the gene expression level, the immunofluorescence analysis demonstrated that this inflammasome could be partially activated by some of the air pollutants with lower co-localization with ASC compared to the other NLR inflammasomes (Figure 4a). This could be explained via the hypothesis that NLRP3 activation in response to air pollutants might be a very fast event, faster than the earliest timepoint analyzed in our study; on the other hand, under noxious stimuli, the cutaneous NLRP3 inflammasome might be subjected to modifications occurring at the post-translational level that affect its activity. Post-translational modification (PTM) events, including ubiquitination, SUMOylation, phosphorylation, etc., can modulate inflammasome activation in many contexts, including in response to air pollutants, resulting in their activation or inhibition [75]. For example, CS exposure can promote NLRP3 inhibition via ubiquitination [76], whereas O_3_ can promote NLRP1 activation via the ubiquitin-proteasome system. Moreover, it is worth mentioning that although colocalization with ASC rarely occurred for almost all the NLR receptors under our experimental conditions, the role of ASC in inflammasome activation is still not completely clear, since it is not always required for the inflammasomes’ activation. For instance, there is evidence demonstrating that since NLRC4 does not display a PYD domain, it can directly interact with caspase 1 without recruiting ASC [77]. NLRP1 can also directly interact with caspase 1 due to the presence of an additional CARD domain in its C-terminus that can directly recruit pro-caspase 1 [78], as well as another inflammatory caspase, caspase 5 [79]. Considering that ASC is predominantly colocalized with NLRP1, it is plausible that the significant increase in the total ASC protein expression levels found in the human skin biopsies in response to air pollutant exposure (Figure 3e) might have been due to NLRP1 activation. Nevertheless, our data showed that, besides NLRP1, NLRP6 was the preferred inflammasome modulated by the environmental pollutants in the skin biopsies. This was the most surprising result, considering that, while NLRP1, NLRP3, and NLRC4 are the most studied inflammasomes in human skin, evidence surrounding the role of NLRP6 in the cutaneous inflammatory response is scarce. NLRP6 is predominantly expressed in the gastrointestinal tract, where it regulates the inflammatory response and the intestinal microbiota in response to infectious stimuli. Interestingly, it has been shown that NLRP6 can upregulate the antimicrobial peptides (AMPs) naturally present in the gut microbiome, protecting it from infectious bacteria [80,81]. Considering that human skin is equipped with AMPs that act as antimicrobial agents, which also regulate the activation of immune cells and the production of pro-inflammatory cytokines [82,83], it is plausible that NLRP6 could modulate cutaneous AMPs in the regulation of the inflammatory response as well. Indeed, a range of evidence shows a strict link between the gut and skin (gut–skin axis), demonstrating an interconnection between the intestinal and cutaneous microbiomes in the manifestation of skin conditions [84]. Of note, increased levels of cutaneous AMPs have been detected in many skin diseases, such as atopic dermatitis and psoriasis [85,86]. Moreover, exposure to O_3_ has been shown to upregulate cutaneous AMPs such as human betadefensins (HBDs) and cathelicidin LL-37 [87], supporting the fact that, in this study, O_3_ showed the greatest effect on NLRP6 modulation both at the gene and protein expression levels. The activation of inflammasomes by pollutant exposure was finally confirmed by the release of the cytokine IL-1β in the media of the human skin biopsies, along with upregulated levels of the pyroptosis mediator Gasdermin D (Figure 5a,b). MP and O_3_ led to the complete activation of the inflammatory pathways after the first period of exposure, as depicted by the increased released levels of IL-1β found in the media of the human skin biopsies, whereas DEE and UV seemed to induce inflammasome activation more in the long term (DAY 4). In addition, exposure to air pollutants promoted the activation of the pyroptosis effector p30 Gasdermin D, whose levels were upregulated both in the short and long term (Figure 5b). Considering that the release of IL-1β found in the media of skin biopsies after the first period of exposure seemed to be particularly high in response to MP and O_3_, as well as the levels of mature Gasdermin D, it is possible that MP and O_3_ represent the main pollutants promoting pyroptosis as an acute response in human skin. Moreover, our results suggest that NLRP1, followed by NLRP6, may represent the preferred inflammasome receptors involved in the MP- and O_3_-induced cutaneous inflammatory response in the short term (DAY 1).

In conclusion, our study shows the different effects that air pollutants may exert on the markers of the oxidative and inflammatory responses and the skin aging process. In particular, as summarized in Appendix A, O_3_ appeared to represent the air pollutant that was most likely to affect inflammasome activation in human skin as an acute response (DAY 1), whereas exposure to UV was found to be the preferred environmental insult able to activate the inflammasomes in the long term (DAY 4). Since UV is also responsible for inducing the greatest damage in terms of DNA oxidation and the activation of MMP-2, it is plausible that the inflammasomes are involved in the establishment of the inflammatory status, contributing to the skin aging process (inflammaging) [25]. Interestingly, among all air pollutants, MP displayed the highest impact in terms of oxidative stress marker induction, such as 4HNE and SESN2, in the short term (DAY 1), along with a tendency to promote an inflammatory status, as displayed by increased levels of IL-1β and the activation of Gasdermin D.

We believe that this is the first study showing the ability of MP to affect the skin’s physiology and it may represent an important modulator of the inflammasome response in human skin. Considering that daily, concomitant exposure to pollutants can synergistically affect the cutaneous oxinflammatory status and compromise the skin barrier function [88], it is plausible that MP can not only exacerbate the air pollutants’ cutaneous damage, but they may more easily penetrate within the *stratum corneum,* thus becoming important noxious stimuli for the skin. For instance, MP exposed to UV were found to be subjected to oxidation or be more prone to release by plastic films, thus becoming even more dangerous to the body [19].

In addition, we also showed for the first time that NLRP6, along with NLRP1, was particularly susceptible to the effect of air pollutants in human skin, thus becoming an important molecular target involved in the inflammatory response of the cutaneous tissue.

## 4. Materials and Methods

### 4.1. Skin Biopsies, Culture, and Treatments

Human skin explants were obtained from elective abdominoplasties and the protocol was approved by the Institutional Biosafety (IBC) Committee at NC State University. Subcutaneous fat was removed from the abdominoplasties and 12 mm punch biopsies were collected and rinsed in PBS. Biopsies were then placed in 6-well plates and cultured in DMEM high-glucose containing 10% FBS, 100 U/mL penicillin, and 100 μg/mL streptomycin at 37 °C in 5% CO_2_. After 24 h of recovery in the incubator, the culture medium was replaced with fresh medium, and the biopsies were exposed to the different air pollutants [88]. Pollutant exposure was performed as follows: 20 µL of a 500 µg/mL microplastics solution (cat. 102 Phosphorex, LLC, Hopkinton, MA, USA, 01748) was first applied to skin biopsies using a glass rod. Skin biopsies were incubated for 24 h and then exposed to 200 mJ UVA/UVB light [88] or 0.25 ppm of O_3_ for 2 h as previously described, 30 min of diesel engine exhaust (DEE) as in reference [89], or 30 min of cigarette smoke. Skin biopsies were exposed every day for 4 days and the samples were collected 24 h after the end of the first air pollutant exposure (DAY 1) or after 4 days (DAY 4). Medium was changed every day before exposure to air pollutants and collected at the indicated timepoints for further analysis.

### 4.2. Immunofluorescence Staining

Human skin samples were fixed in 10% neutral buffered formalin for 24 h RT and dehydrated in increasing concentrations of alcohol (70, 80, 90, and 100%) for 30 min and then in xylene for 45 min, twice, using an automated Leica tissue processor (Leica Biosystems, Deer Park, IL, USA). The samples were left in paraffin overnight and embedded in paraffin to create blocks. For immunohistochemical analysis, 4 μm sections of skin biopsies were cut using a microtome (Leica Biosystems, Deer Park, IL, USA) and then deparaffinized in xylene and rehydrated in a decreasing alcohol gradient as previously described [88]. The sections were subjected to heat-based antigen retrieval using a sodium citrate buffer (C9999, Sigma-Aldrich, Merck Millipore, Burlington, MA, USA) (pH 6.0) at a sub-boiling temperature of 95 °C in a water bath for 10 min. Samples were cooled down to room temperature for 25 min, washed twice in PBS and then blocked with 5% BSA in PBS for 45 min at room temperature. Sections were then incubated overnight with primary antibodies at 4 °C as follows: NLRP1 (sc-166368Santa Cruz Biotechnology Inc., Dallas, TX, USA), NLRP3 (cat. NBP1-97601, Novus Biological, Littleton, CO, USA) 1:100, NLRP6 (NBP2-31372, Novus Biological, Littleton, CO, USA) 1:200, NLRC4 (NB100-561425, Novus Biological, Littleton, CO, USA) 1:200, ASC (cat. NBP1-78977, Novus Biological, Littleton, CO, USA) 1:150 or ASC (sc-271054, Santa Cruz Biotechnology Inc., Dallas, TX, USA) 1:50, 8-OHdG (sc-393871, Santa Cruz Biotechnology Inc., Dallas, TX, USA), type I collagen (ab-138492, abcam, Cambridge, UK), Sestrin 2 (cat. 10795-1-AP, Proteintech, Rosemont, IL, USA), 4HNE (AB5605, MilliporeSigma, Burlington, MA, USA) PBS, BSA 0.25%. The day after, samples were washed in PBS 3 times for 5 min and incubated with fluorochrome-conjugated secondary antibodies (dil. 1:1000) (Alexa Fluor 568 A11004 or Alexa Fluor 488 A11055) in 0.25% BSA in PBS at RT. Nuclei were stained with DAPI (D1306 Invitrogen, Thermo-Fisher Scientific, Waltham, MA, USA). Coverslips were mounted onto glass slides using Fluoromount-G™ Mounting Medium (00-4958-02, ThermoFisher Scientific, Waltham, MA, USA), and examined using a Zeiss Z1 AxioObserver LSM10 confocal microscope at 40× magnification. Images were quantified using the ImageJ software 1.53a (Java 1.8.0_172, National Institutes of Health, Bethesda, MD, USA).

### 4.3. RNA Extraction and Quantitative Real-Time PCR (rt-PCR)

RNA extraction from human skin biopsies was performed using TRIzol™ Reagent (Cat. 15596026, Invitrogen, ThermoFisher Scientific, Waltham, MA, USA) as previously described in [87]. Briefly, skin samples were placed in Precellys Eppendorf tubes containing metallic beads (cat. 15-340-151, Fisherbrand, ThermoFisher Scientific, Waltham, MA, USA) and 700 μL of Trizol was added to the samples. Tissues were then homogenized using a TissueLyser (Qiagen, Germantown, MD, USA) equipped with liquid nitrogen, by performing 3 cycles of 6500 rpm for 30 s each. Samples were then subjected to a modified version of the phenol–chloroform extraction protocol described previously [6]. Briefly, 140 µL RNase-free chloroform was added to each tube and then centrifuged for 15 min at 12,000× *g* at 4 °C. The aqueous phase containing RNA was transferred to a new tube and the chloroform step was repeated. RNA precipitations were assessed by adding 350 µL of RNase-free isopropanol and centrifuging at 4 °C, 12,000× *g* for 10 min. The RNA pellet was washed 3 times in 75% EtOH by centrifuging the samples at 4 °C, 10,000× *g* for 10 min. The pellet was then suspended in 35 µL of nuclease-free water and quantified using a Nanodrop spectrophotometer (ThermoFisher Scientific, Waltham, MA, USA). Then, 1 μg of total RNA from each sample was reverse transcribed in cDNA using the iScript cDNA Synthesis Kit (cat. 1708841, Bio-Rad Laboratories, Inc., Hercules, CA, USA), according to the manufacturer’s instructions. The mRNA levels of NLRP1, NLRP3, NLRP6, ASC, and IL-1β were analyzed by performing a quantitative real-time PCR using the SsoAdvanced Universal SYBR Green Supermix (cat. 1725271, Bio-Rad Laboratories, Inc., Hercules, CA, USA), on a Roche LightCycler 480 machine, according to the manufacturer’s protocol.

Gene expression was quantified based on the number of cycles necessary to reach a predetermined threshold value (Ct value) for each sample, and glyceraldehyde 3-phosphate dehydrogenase (*GAPDH*) was employed as the reference gene. After normalization, the fold change was determined using the 2^−ΔΔCT^ method. The primer sequences are listed in the Table 1.

### 4.4. Protein Extraction and Western Blotting

Skin explants were placed in Precellys Eppendorf tubes containing metallic beads (cat. 15-340-151, Fisherbrand, ThermoFisher Scientific, Waltham, MA, USA) and homogenized as previously described [88]. Briefly, 350 µL of T-PER™ Tissue Protein Extraction Reagent (cat. 78510, ThermoFisher Scientific, Waltham, MA, USA) containing Halt™ Protease Inhibitor Cocktail (100×) (cat. 78429 ThermoFisher Scientific, Waltham, MA, USA) was added to each sample and tissues were subjected to 3 cycles of 6500 rpm, 30 s each, using the TissueLyser (Qiagen, Germantown, MD, USA) equipped with liquid nitrogen. Samples were then centrifuged at 16,000× *g* rpm, 30 min at 4 °C, and the protein content was then measured using Bradford assays and equivalent amounts of proteins were loaded onto polyacrylamide SDS gels. Gels were electroblotted onto nitrocellulose membranes (0.45 µm pore size) (cat. 1620115, Bio-Rad Laboratories, Inc., Hercules, CA, USA) and blocked in PBS-Tween 0.5% with 5% non-fat dry milk. The following primary antibodies diluted in PBS-T and milk 1% were used to incubate the membranes overnight at 4 °C: ASC (cat. NBP1-78977, Novus Biological, Littleton, CO, USA) 1:1000 and Gasdermin D (sc-393581, Santa Cruz Biotechnology Inc., Dallas, TX, USA) 1:500.

The day after, the membranes were washed in PBS-T and incubated with horseradish peroxidase-conjugated secondary antibodies (cat. 170–6515, 170–6516, Bio-Rad Laboratories, Inc., Hercules, CA, USA) diluted 1:5,000 in TBS-T with 1% non-fat milk for 90 min at RT. Bound antibodies were detected by chemiluminescence via the Clarity Western ECL Substrate (cat. 1705061, Bio-Rad Laboratories, Inc., Hercules, CA, USA) using the ChemiDoc Imaging System (Bio-Rad Laboratories, Hercules, CA, USA). β-actin (A3854 Sigma-Aldrich, Burlington, VT, USA) was used as a loading control at a 1:50,000 dilution in TBS-T with 1% non-fat milk. Densitometry analysis was performed using the Image J software 1.53a (Java 1.8.0_172, National Institutes of Health, Bethesda, MD, USA).

### 4.5. MMP-2 Zymography

The protein content of media samples was quantified using Bradford assays. First, 10 µg of the media sample was loaded into an 8% acrylamide 1 mm gel and run at 125 V for 90 min. Gels were carefully removed and placed in washing buffer (2.5% Triton X-100, 50 mM Tris HCl, 5 mM CaCl_2_, H_2_O) for 30 min at room temperature with gentle agitation twice. The washing buffer was then decanted and rinsed with 100 mL of double-distilled water. The gel was incubated for 30 min in 100 mL of developing buffer (50 mM Tris-HCl, 5 mM CaCl_2_, 200 mM NaCl, H_2_O) with gentle agitation; the solution was then decanted and replaced with fresh developing buffer overnight at room temperature. The next day, the developing buffer was removed, and the gel was stained with 50 mL of staining solution (0.1% Coomassie Blue) for 1 h with agitation. Following staining, the gel was washed with double-distilled water until excess staining solution was removed. The gels were then washed with destaining solution (50% H_2_O, 40% methanol, 10% acetic acid) until gelatinolytic activity was evident. Images were then taken with the ChemiDoc Imaging System (Bio-Rad Laboratories, Hercules, CA, USA) and bands were quantified using the ImageJ software 1.53a (Java 1.8.0_172, National Institutes of Health, Bethesda, MD, USA).

### 4.6. ELISA for IL-1β

IL-1β levels were measured in the media of skin biopsies collected at the indicated timepoints using the IL-1β ELISA kit (cat. DY201-05, Novus Biologicals, Centennial, CO 80112, USA), according to the manufacturer’s instructions. The absorbance was measured with a spectrophotometer equipped with a filter of 450 nm, using 570 nm as a reference wavelength. IL-1β levels were expressed as pg/mL in culture media according to the manufacturer’s instructions, and the Gen5 2.0 software (BioTek, Agilent, Santa Clara, CA, USA) was used for the detection.

### 4.7. Statistical Analysis

The statistical analysis was performed using GraphPad Prism 9 (Version 9.4.1 (458), GraphPad Software Inc., La Jolla, CA, USA) with an analysis of variance (1-way or 2-way ANOVA), followed by Tukey’s post-hoc test, for each of the variables tested. Data are expressed as the mean ± SD of duplicate determinations obtained in three independent experiments, and statistical significance was considered at *p* < 0.05. For all experiments, control values were set to 1.0 and other values expressed as a fold change.

## Figures and Tables

**Figure 1 ijms-24-16674-f001:**
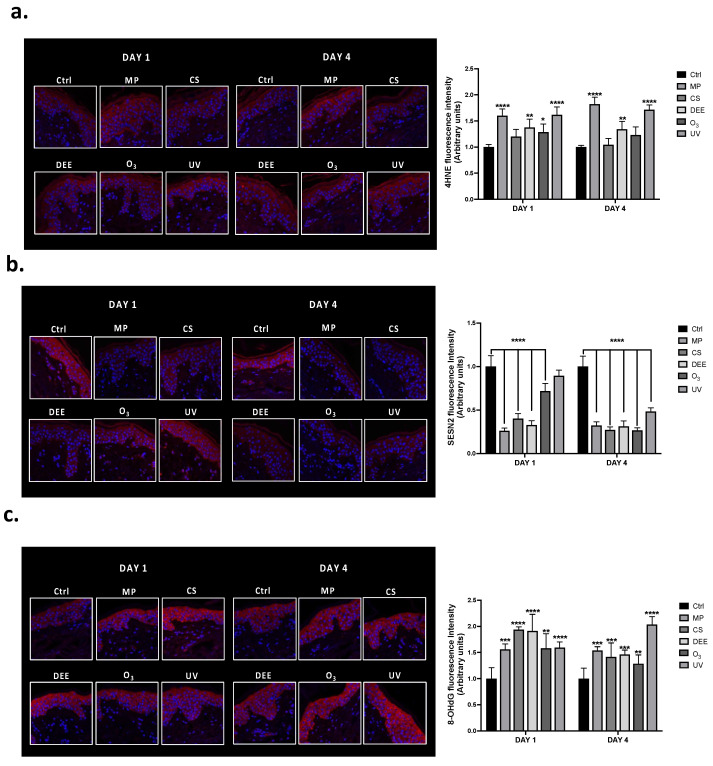
Immunofluorescence staining for 4HNE (**a**), SESN2 (**b**), and 8-OHdG (**c**) in human skin biopsies exposed to different air pollutants and collected at the indicated timepoints (DAY 1 and DAY 4). Red represents 4HNE (**a**), SESN2 (**b**), 8-OHdG (**c**) and the blue staining (DAPI) represents nuclei. Images were taken at 40× magnification and the fluorescent signal was quantified using ImageJ software 1.53a (Java 1.8.0_172). Data are the results of the averages of at least three different experiments, * *p* < 0.05; ** *p* < 0.005; *** *p* < 0.001; **** *p* < 0.0001 pollutants vs. Ctrl at the respective timepoints (DAY 1 and DAY 4) by 2-way ANOVA followed by Tukey’s post-hoc comparison test.

**Figure 2 ijms-24-16674-f002:**
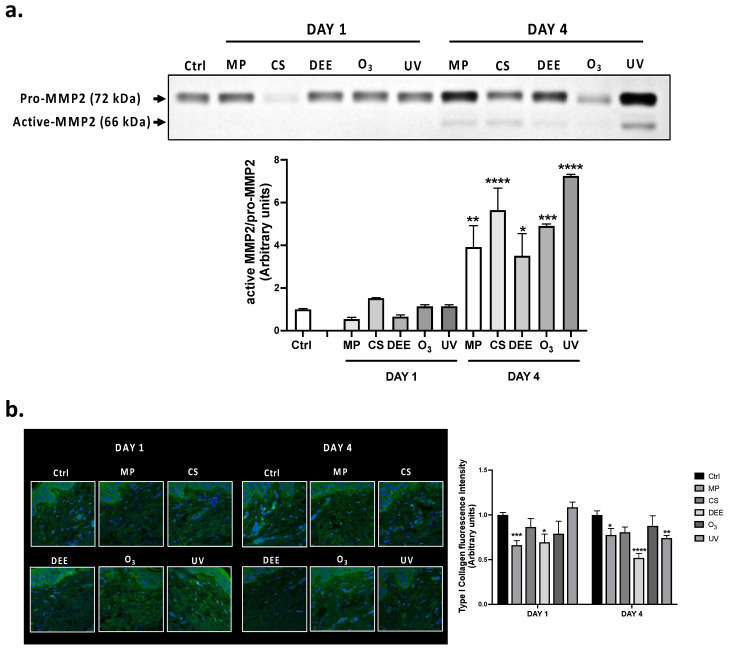
(**a**) Gelatin zymography for MMP-2 in human skin biopsies. The assay was performed on whole skin homogenates (10 µg) collected 1 day and 4 days after air pollutants exposure. The upper panel is representative of 3 independent experiments whereas the underneath graph represents the quantification of active-MMP2 over total pro-MMP-2 assessed by using ImageJ software 1.53a (Java 1.8.0_172). (**b**) Immunofluorescence staining for Type I Collagen in human skin biopsies exposed to air pollutants and collected at the indicated timepoints. Green staining represents Type I Collagen, and the blue staining (DAPI) represents nuclei. Images were taken at 40× magnification and the fluorescent signal was quantified using ImageJ software 1.53a (Java 1.8.0_172). Data are the results of the averages of at least three different experiments, * *p* < 0.05; ** *p* < 0.005; *** *p* < 0.001; **** *p* < 0.0001 pollutants vs. Ctrl at the respective timepoints (DAY 1 and DAY 4) by 2-way ANOVA followed by Tukey’s post-hoc comparison test.

**Figure 3 ijms-24-16674-f003:**
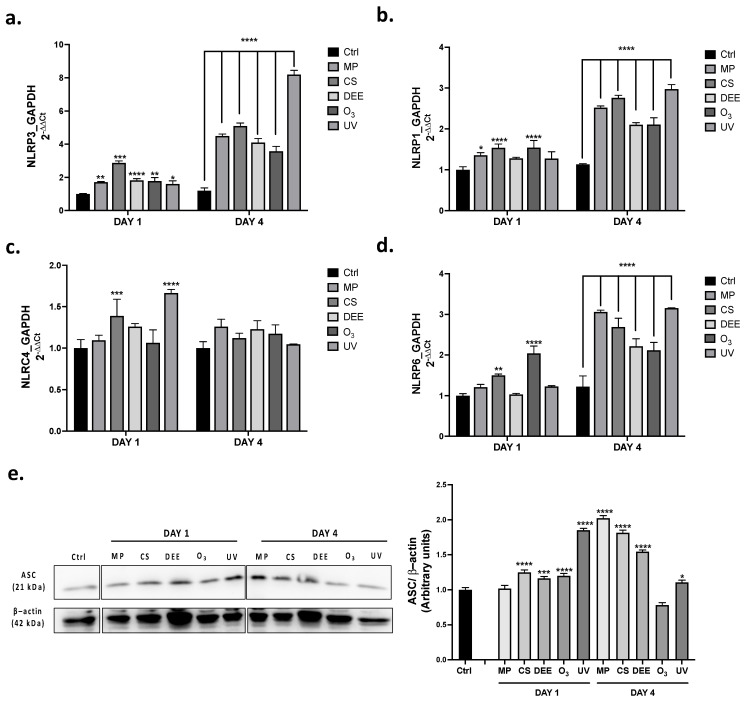
mRNA expression levels of NLRP3 (**a**), NLRP1 (**b**), NLRC4 (**c**) and NLRP6 (**d**) in human skin biopsies exposed to air pollutants and collected at DAY 1, DAY 2 and DAY 4 post exposure. (**e**) Protein expression levels of ASC in human skin biopsies exposed to air pollutants and analyzed at the indicated timepoints. The protein expression level was quantified using ImageJ software 1.53a (Java 1.8.0_172) and β-actin was used as internal control. Data are the results of the averages of at least three different experiments, * *p* < 0.05; ** *p* < 0.005; *** *p* < 0.001; **** *p* < 0.0001 pollutants vs. Ctrl at the respective timepoints (DAY 1 and DAY 4) by 2-way ANOVA followed by Tukey’s post-hoc comparison test.

**Figure 4 ijms-24-16674-f004:**
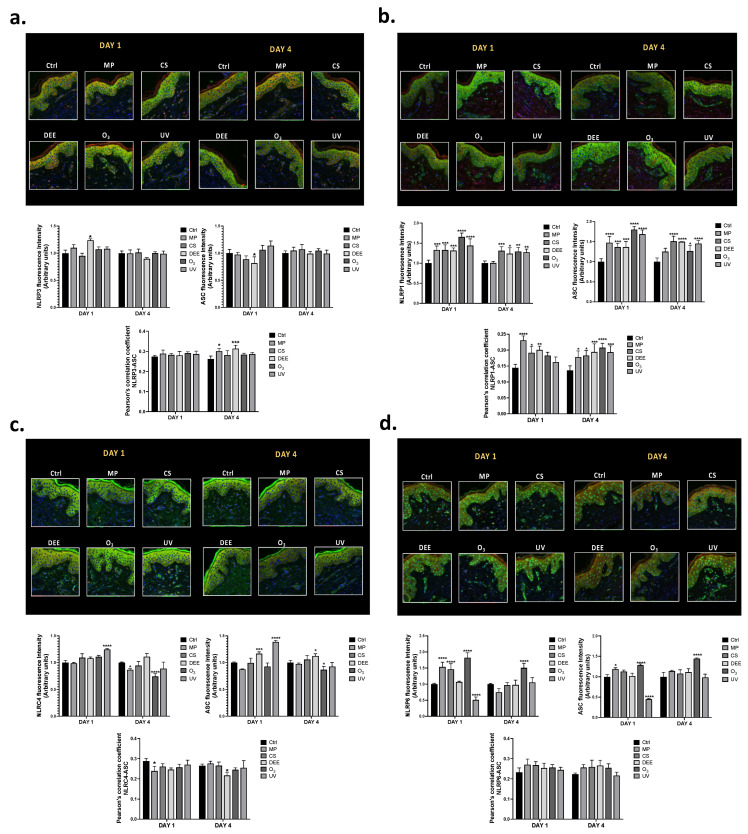
Immunofluorescence staining for NLRP3 (green staining) and ASC (red staining) (**a**); NLRP1 (red staining) and ASC (green staining) (**b**); NLRC4 (green staining) and ASC (red staining) (**c**); and NLRP6 (green staining) and ASC (red staining) (**d**) in human skin biopsies exposed to air pollutants and collected at the indicated timepoints. Blue staining (DAPI) represents nuclei. Images were taken at 40× magnification and the fluorescent signal was quantified using the ImageJ software 1.53a (Java 1.8.0_172). The colocalization between NLRP3, NLRP1, NLRC4, or NLRP6 and ASC is expressed as Pearson’s correlation coefficient. Data are the results of the averages of at least three different experiments, * *p* < 0.05; ** *p* < 0.005; *** *p* < 0.001; **** *p* < 0.0001 pollutants vs. Ctrl at the respective timepoints (DAY 1 and DAY 4) by 2-way ANOVA followed by Tukey’s post-hoc comparison test.

**Figure 5 ijms-24-16674-f005:**
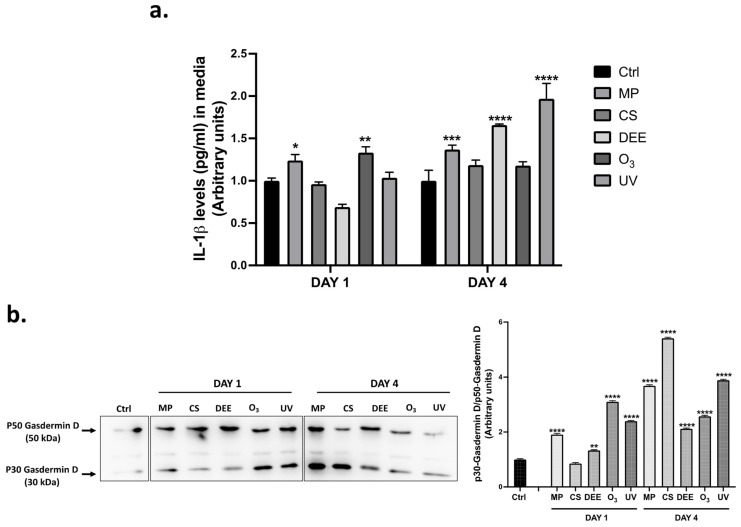
(**a**) Released levels of IL-1β expressed as pg/mL in media of human skin biopsies exposed to air pollutants and analyzed at DAY 1 and DAY 4. (**b**) Protein expression levels of p30 Gasdermin D over p50 Gasdermin D in human skin biopsies exposed to air pollutants and analyzed at the indicated timepoints. Protein expression level was quantified using the ImageJ software 1.53a (Java 1.8.0_172) and β-actin was used as an internal control. Data are the result of the averages of at least three different experiments, * *p* < 0.05; ** *p* < 0.005; *** *p* < 0.001; **** *p* < 0.0001 pollutants vs. Ctrl at the respective timepoints (DAY 1 and DAY 4) by 2-way ANOVA followed by Tukey’s post-hoc comparison test.

**Table 1 ijms-24-16674-t001:** Primers sequences.

Gene	Forward Sequence	Reverse Sequence
*NLRP1*	ACCCTCTTAACTCCGGGACA	GAGTGCGCTTTATTGGCGAG
*NLRP3*	CGGGGCCTCTTTTCAGTTCT	CCCCAACCACAATCTCCGAA
*NLRP6*	CCTGTGAAGGAATCACCTCTCT	GTCCATGGGGTCTCTTCCTCC
*ASC*	ATGCGCTGGAGAACCTGA	TCTCCAGGTAGAAGCTGACCA
*IL-1β*	CACGATGCACCTGTACGATCA	GTTGCTCCATATCCTGTCCCT
*GAPDH*	TCGGAGTCAACGGATTTGGT	TTCCCGTTCTCAGCCTTGAC

## Data Availability

The authors confirm that all relevant data are available from the corresponding author, upon request.

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
