# Peer review of "Comparison of Pollutant Effects on Cutaneous Inflammasomes Activation"

_ijms, 2023, doi:10.3390/ijms242316674_

Round 1

Reviewer 1 Report

Comments and Suggestions for Authors

The manuscript: „Comparison of pollutants effect on cutaneous inflammasomes activation” by Ivarsson et al. provides evidence that exposure of human skin to various environmental pollutants and factors induces oxidative stress and an inflammatory response that is accompanied by oxidative damage to biomolecules.

In “Introduction” the authors give precise and useful information on their subject. The manuscript is interesting; the design and statistical analysis is good. The discussion adequately explains experimental results.

The conclusion is clear and summarizes the complex effects of various environmental factors on human skin, pointing to potential molecular mechanisms related to the induction of an oxinflammatory response.

In addition, the used literature adequately represents the current knowledge of all scientific “hot points” which are the subjects of the manuscript.

I recommend publishing the manuscript in this form with the suggestion to correct minor typos.

Minor typos:

Abstract:

"Diesel exhaust (DEE)" - is DEE an abbreviation for Diesel exhaust, or "diesel exhaust engine". Please check and correct through the manuscript.

“…and UV represented…” - UV or UVR. Please check and correct through the manuscript.

“Keywords:” - Please check. I did not find Keywords.

Introduction:

“…superoxide radical anion (O2-)…” - Please write the chemical symbol for the superoxide anion radical correctly, with a dot in the superscript.

“leading to the production of ROS or nonradical species such as aldehydes (4-hydroxy-nonenal, 4HNE)...” - Please correct the sentence to avoid confusion because ROS includes nonradical species.

“(i.e. apoptosis)21,22, “human skin 23” - Please synchronize the citations in the text.

“Although inflammasomes can be activated by a variety of stimuli, reactive oxygen species (ROS) have been addressed as one the main responsible insults of inflammasomes activation 24.” - Please introduce the abbreviation the first time it appears.

“…Diesel Exhaust (DEE)…” - Please check and correct through the manuscript.

“…sample and GAPDH was employed…” - Please enter full name for GAPDH.

“…involved in ECM degradation…” - Please enter full name for ECM.

Discussion:

“Air pollutants and UV have…” - UV or UVR. Please check and correct through the manuscript.

“…levels of 4-hydroxynonenal (4HNE)…” - The abbreviation is already introduced.

“Other pollutants such as cigarette smoke (CS)…” - The abbreviation is already introduced. Please check and correct through the manuscript.

”…and NRF2 transcription factors…” - Please enter full name for NRF2.

“…particulate matter (PM)…” - The abbreviation is already introduced.

Author Response

Reviewer 1

Comments and Suggestions for Authors

The manuscript: “Comparison of pollutants effect on cutaneous inflammasomes activation” by Ivarsson et al. provides evidence that exposure of human skin to various environmental pollutants and factors induces oxidative stress and an inflammatory response that is accompanied by oxidative damage to biomolecules.

In “Introduction” the authors give precise and useful information on their subject. The manuscript is interesting; the design and statistical analysis is good. The discussion adequately explains experimental results.

The conclusion is clear and summarizes the complex effects of various environmental factors on human skin, pointing to potential molecular mechanisms related to the induction of an oxinflammatory response.

In addition, the used literature adequately represents the current knowledge of all scientific “hot points” which are the subjects of the manuscript.

I recommend publishing the manuscript in this form with the suggestion to correct minor typos. 

Minor typos:

Abstract:

Point 1

"Diesel exhaust (DEE)" - is DEE an abbreviation for Diesel exhaust, or "diesel exhaust engine". Please check and correct through the manuscript.

Answer 1

The typo has been corrected and the correction has been highlighted throughout the manuscript.

Point 2

“…and UV represented…” - UV or UVR. Please check and correct through the manuscript.

Answer 2

The instances of UVR have been removed and replaced with UV to make abbreviation consisted throughout the paper.

Point 3

“Keywords:” - Please check. I did not find Keywords.

Answer 3

The appropriate keywords have been added to the manuscript and highlighted.

Introduction:

Point 4

“…superoxide radical anion (O2-)…” - Please write the chemical symbol for the superoxide anion radical correctly, with a dot in the superscript.

Answer 4

The chemical symbol has been corrected as requested in the manuscript. The correction has been highlighted in the text.

Point 5

“leading to the production of ROS or nonradical species such as aldehydes (4-hydroxy-nonenal, 4HNE)...” - Please correct the sentence to avoid confusion because ROS includes nonradical species.

Answer 5

 “or” has been replaced with “and other” to avoid any confusion. The correction has been highlighted in the manuscript.

Point 6

“(i.e. apoptosis)21,22, “human skin 23” - Please synchronize the citations in the text.

Answer 6

The citations have been synchronized in the manuscript as requested.

Point 7

“Although inflammasomes can be activated by a variety of stimuli, reactive oxygen species (ROS) have been addressed as one the main responsible insults of inflammasomes activation 24.” - Please introduce the abbreviation the first time it appears.

Answer 7

The abbreviation has been introduced the first time it appears in the manuscript, as requested. The correction has been highlighted.

Point 8

“…Diesel Exhaust (DEE)…” - Please check and correct through the manuscript.

Answer 8

The correction has been made throughout manuscript, first mention: Diesel engine exhaust, subsequent: DEE.

Point 9

“…sample and GAPDH was employed…” - Please enter full name for GAPDH.

Answer 9

The full name for GAPDH has been added as requested in the  paragraph 2.3 of material and methods. The correction has been highlighted.

Point 10

“…involved in ECM degradation…” - Please enter full name for ECM.

Answer 10

The full name for ECM has been added in the paragraph 3.2 of the results section. The correction has been highlighted.

Discussion:

Point 11

“Air pollutants and UV have…” - UV or UVR. Please check and correct through the manuscript.

Answer 11

The instances of UVR have been removed and replaced with UV to make abbreviation consisted throughout the paper.

Point 12

“…levels of 4-hydroxynonenal (4HNE)…” - The abbreviation is already introduced.

Answer 12

“4-hydroxynonenal” has been deleted as requested and replaced with the acronym.

Point 13

“Other pollutants such as cigarette smoke (CS)…” - The abbreviation is already introduced. Please check and correct through the manuscript.

Answer 13

The manuscript has been updated accordingly.  

Point 14

”…and NRF2 transcription factors…” - Please enter full name for NRF2.

Answer 14

The full name for NRF2 has been entered in the discussion section of the manuscript and the correction has been highlighted.

Point 15

“…particulate matter (PM)…” - The abbreviation is already introduced.

Answer 15

The manuscript has been revised as requested.

Reviewer 2 Report

Comments and Suggestions for Authors

The study conducted by Ivarsson et al have conducted the study using human skin biopsies that were exposed to pollutants such as UV radiation, Diesel Exhaust, Cigarette Smoke, Ozone and Microparticles. The read out were markers related to oxidative stress, DNA damage and skin aging. The novelty of the study in elucidation of NLR inflammasomes, particularly the NLRP1 leading to various effector functions such as scaffold formation within ASC, release of inflammatory cytokines and pyroptosis. 

Experimental approach and presentation of results is clear. Discussion is long and extensive yet authors have refrained from proposing how these findings could be relevant for actual clinical manifestation on skin. They have not cited any clinical/observational studies that reported effects of exposure to pollutants. 

Author Response

Reviewer 2

Comments and Suggestions for Authors

The study conducted by Ivarsson et al have conducted the study using human skin biopsies that were exposed to pollutants such as UV radiation, Diesel Exhaust, Cigarette Smoke, Ozone and Microparticles. The read out were markers related to oxidative stress, DNA damage and skin aging. The novelty of the study in elucidation of NLR inflammasomes, particularly the NLRP1 leading to various effector functions such as scaffold formation within ASC, release of inflammatory cytokines and pyroptosis. 

Point 1

Experimental approach and presentation of results is clear. Discussion is long and extensive yet authors have refrained from proposing how these findings could be relevant for actual clinical manifestation on skin. They have not cited any clinical/observational studies that reported effects of exposure to pollutants. 

Answer 1

We thank the reviewer for the suggestion. Clinical/ observational studies have been cited in the discussion section and discussed. All the additional parts have been highlighted in the manuscript.